# Iron and Nitrogen-Doped Wheat Straw Hierarchical Porous Carbon Materials for Supercapacitors

**DOI:** 10.3390/nano14211692

**Published:** 2024-10-23

**Authors:** Xiaoshuai Sun, Xiangyu Chen, Jiahua Ma, Chuanshan Zhao, Jiehua Li, Hui Li

**Affiliations:** State Key Laboratory of Biobased Material and Green Papermaking, Qilu University of Technology, Shandong Academy of Sciences, Jinan 250353, China; 17861407148@163.com (X.S.); colacxy@163.com (X.C.); 13290394238@163.com (J.M.); lijiehua1995@163.com (J.L.); 15106900618@163.com (H.L.)

**Keywords:** carbonization, energy storage, Fe/N co-doped hierarchical porous carbon, specific capacity, supercapacitor electrode, wheat straw

## Abstract

In this paper, we prepared a new type of iron and nitrogen co-doped porous carbon material (WSC-Fe/N) using a carbonization–activation process with wheat straw as a precursor and FeCl_3_ and NH_4_Cl as co-doping agents and analyzed the electrochemical properties of the resulting electrode material. Through precise control of the doping elements and carbonization temperature (900 °C), the resulting WSC-Fe/N-900 material exhibits abundant micropores, uniform mesopores, a significant specific surface area (2576.6 m^2^ g^−^^1^), an optimal level of iron doping (1.7 wt.%), and excellent graphitization. These characteristics were confirmed through X-ray diffraction and Raman spectroscopy. Additionally, the WSC-Fe/N-900 electrode demonstrated a specific capacitance of 400.5 F g^−^^1^ at a current density of 0.5 A g^−^^1^, maintaining a high capacitance of 308 F g^−^^1^ even at 10 A g^−^^1^. The solid-state symmetric supercapacitor in an aqueous electrolyte achieved an energy density of 9.2 Wh kg^−^^1^ at a power density of 250 W kg^−^^1^ and maintained an energy density of 6.5 Wh kg^−^^1^ at a power density of 5000 W kg^−^^1^, demonstrating remarkable synergistic energy–power output characteristics. In terms of structural properties, the porous characteristics of WSC-Fe/N-900 not only enhance the specific surface area of the electrode but also improve the diffusion capability of electrolyte ions within the electrode, thereby enhancing capacitance performance. The reliability of the electrode material demonstrated good performance in long-term cycling tests, maintaining a capacitance retention rate of 93% after 10,000 charge–discharge cycles, indicating excellent electrochemical stability. Furthermore, over time, the aging effect of the WSC-Fe/N-900 electrode material is minimal, maintaining high electrochemical performance even after prolonged use, suggesting that this material is suitable for long-term energy storage applications. This study introduces a novel strategy for producing porous carbon materials for supercapacitors, advancing the development of economically efficient and environmentally friendly energy storage solutions.

## 1. Introduction

Researchers from all over the world have been concentrating more on the development of green and renewable energy storage technologies in response to the global energy problem and environmental pollution. Due to their high power density, rapid charge and discharge capabilities, and extended cycle life, supercapacitors have found widespread application in energy storage for electric vehicles and portable electronic devices [1,2,3]. Supercapacitors are a sophisticated kind of energy storage technology that fills the vacuum left by batteries and conventional capacitors. They go by the names electric double-layer capacitors and electrochemical capacitors as well [4,5]. However, a limiting factor of conventional supercapacitors is that they use electrode materials which are typically costly and require labor-intensive preparation procedures, such as graphene, carbon nanotubes, and activated carbon [6,7,8]. Thus, the quest for affordable, eco-friendly, and high-performing electrode materials has emerged as a prominent area of research [9,10,11].

The resource utilization of agricultural waste provides a new solution to this problem [12]. In particular, wheat straw is an abundant and cheap form of agricultural waste generated in large quantities after annual crop harvests that is usually directly burned or discarded, resulting in the waste of valuable resources and pollution of the environment [13,14]. The conversion of wheat straw into carbon materials obtained from biomass can not only help achieve efficient waste usage, but also reduce environmental pollution and encourage resource recycling [15,16].

When subjected to appropriate carbonization and activation processes, wheat straw—an abundant carbon source—can be converted into biomass-derived carbon composites with an excellent specific surface area and an optimized pore structure [17]. Because of these qualities, wheat straw-derived carbon (WSC) has exceptional electrochemical performance, which makes it a perfect electrode material for supercapacitors [18,19]. Notably, under high-temperature carbonization, wheat straw can be transformed into precursor carbon compounds. Subsequent chemical activation (through the use of potassium hydroxide, phosphoric acid, or other activators) can further improve its porosity and specific surface area, thereby significantly enhancing its electrochemical performance. The resulting wheat straw-derived carbon materials exhibit substantial charge storage capacity and energy density as supercapacitors [20].

To this end, this study aimed to propose a method for preparing carbon materials derived from wheat straw biomass for application in supercapacitors. In this study, we introduce a simple and effective chemical activation technique to synthesize iron and nitrogen co-doped porous carbon material from wheat straw (WSC-Fe/N). This study presents a novel material for the manufacture of supercapacitors, which supports green energy storage technologies and offers a viable approach for the resource utilization of agricultural waste. Furthermore, this research advances the goal of developing economical, environmentally friendly, and efficient energy storage solutions, providing robust technical support for the future of sustainable energy systems.

## 2. Materials and Methods

### 2.1. Conversion of Wheat Straw to WSC-Fe/N

To convert wheat straw into WSC-Fe/N, the wheat straw was first cut into approximately 5 cm long pieces and thoroughly washed with deionized water. The cleaned straw segments were dried in a vacuum oven at 70 °C for more than 48 h. Subsequently, 200 mL of aqueous solution was combined with 5 g of chopped wheat straw and 5 g of potassium hydroxide (KOH), and the resulting solution was stirred with a magnetic stirrer for 2 h. Washing with deionized water and hydrochloric acid (HCl) eliminated any remaining KOH. The mixture was then transferred to a vacuum oven set at 110 °C until a solid mass formed. Then, 2.5 g of ferric chloride (FeCl_3_) was added to 5 g of the dried solid mixture in 200 mL of aqueous solution and stirred for 3 h. The resulting solution was again placed in a vacuum oven at 110 °C until a solid mixture was obtained. Subsequently, 1 g of the solid mixture was thoroughly ground with 20 g of ammonium chloride (NH_4_Cl). The powdered material was placed into a crucible and subsequently transferred into a ceramic boat, which was positioned inside a tube furnace. The mixture was heated under a nitrogen atmosphere at a rate of 5 °C min^−1^ to a temperature range of 800 °C–1000 °C and maintained at these temperatures for 2 h. After cooling to room temperature, contaminants were removed from the resulting products by repeatedly washing with distilled water and a 1.5 M HCl solution. The prepared product was placed in a vacuum oven at 70 °C and dried for over 24 h. The obtained test specimens were labeled as WSC-Fe/N. Figure 1 illustrates the complete preparation process.

### 2.2. Fabrication of Supercapacitor

The electrode preparation process was performed as follows. The synthesized sample was combined with polytetrafluoroethylene and conductive carbon black in an 8:1:1 mass ratio in ethanol. The solution was then heated to 60 °C and stirred to promote the evaporation of ethanol, yielding a homogeneous slurry. In the assessment of the three-electrode system, platinum foil was utilized as the counter electrode, a Hg/HgO electrode acted as the reference electrode, and nickel foam was used as the current collector. The electrolyte employed for the experiments was a 6 M KOH solution.

For the preparation of the working electrode, the slurry was uniformly applied onto a current collector and compressed at 10 MPa for 30 s, resulting in an active material loading of approximately 2 mg per electrode. Nickel foam was sectioned into discs with a diameter of 12 mm for use in the two-electrode system test. The slurry was uniformly applied to the circular nickel foam discs, and the working electrode was prepared by pressing the coated foam at 10 MPa for 30 s, resulting in an active material loading of approximately 3 mg per electrode.

For symmetrical supercapacitor testing, a 2025 coin cell was made using two electrodes with equal active material loading as the positive and negative electrodes. A cellulose-based membrane (NKK TF4030, Head Office, Kochi, Japan) was used to separate the electrodes, with a 6 M KOH electrolyte system serving in the role of the electrolyte. The PARSTAT 2273 electrochemical workstation was employed to record cyclic voltammetry (CV) and electrochemical impedance spectroscopy, while the LAND CT 2001A instrument was utilized for measuring constant current charge–discharge curves.

### 2.3. Structural Characterization of WSC-Fe/N

A transmission electron microscope (TEM, JEM-2100, Tokyo, Japan, JEOL Ltd.) and a scanning electron microscope (SEM, Zeiss EVO−18, 10 kV, Jena, Germany, Carl Zeiss AG) were used to analyze the microstructural features of the samples. Before SEM analysis, the samples were gold-coated to enhance conductivity. For TEM analysis, the samples were prepared using microgrid copper meshes. The surface area and pore structure of the materials were measured using a surface area and pore size analyzer (BET, Kubo X1000, Beijing, China, Beijing Builder Electronic Technology Co., Ltd.), with pore size distribution analysis conducted according to the Horvath–Kawazoe and Barrett–Joyner–Halenda models. X-ray diffraction (XRD, Bruker D8, Cu-Kα) was employed to characterize the crystalline structure of the test specimens, with a scan rate set at 10° min^−^^1^ over a range of 2θ = 5–80°. Additionally, the chemical states and elemental composition of the materials were analyzed using X-ray photoelectron spectroscopy (XPS, Axis Ultra DLD, Kyoto, Japan, AXIS Supra+ − Shimadzu Corporation). Raman spectroscopy (LabRAM Aramis, Palaiseau, France, HORIBA Jobin Yvon) was employed to assess the degree of disorder in the carbon materials, with a test range of 400–2200 cm^−^^1^.

## 3. Results and Discussion

### 3.1. Structural Characterization and Analysis

#### 3.1.1. Morphology Characterization

Changes in the pore structures of the porous carbon samples subjected to different carbonization conditions were examined in the SEM images. As shown in Figure 2, the WSC samples exhibited noticeable differences in their structural features. Initially, the untreated WSC sample surface exhibited almost no visible pores. However, prior to the two-step carbonization at 800 °C, 900 °C, and 1000 °C, iron and nitrogen were co-doped into the material. The resulting WSC-Fe/N-T samples (where T represents the different temperatures) exhibited a markedly different morphology compared to the original samples, featuring a much rougher surface and increased porosity. This transformation occurs because, during the carbonization process, NH_4_Cl first decomposes into NH_3_ and HCl at relatively low pyrolysis temperatures. As the pyrolysis temperature increases, the generated NH_3_ further etches the carbon material’s surface, creating micropores [21]. As shown in Figure 2a–c, the reaction intensified to a considerable degree when the pyrolysis temperature reached 900 °C, leading to the creation of noticeable nanopores on the surface of the carbon material.

The WSC-Fe/N-900 sample exhibited the greatest surface roughness and porosity due to its three-dimensional, cheese-like hierarchical porous structure, excellent pore structure, and numerous interconnected pores evenly dispersed over the surface. High porosity implies more active sites and a larger surface area, forming a porous network that is favorable for ion transport [22]. In contrast, the WSC-Fe/N-800 and WSC-Fe/N-1000 samples (Figure 2d) exhibited high porosity but had surface cracks and an uneven pore size distribution, with some micropores converting to mesopores. This disordered structure significantly impacts the performance of electrodes.

Additionally, TEM was used to characterize the sample morphology. Numerous micropores and mesopores were present in the WSC-Fe/N-900 sample, together with a porous nanosheet shape that confers an excellent pore structure, specific surface area, dense pores, and efficient ion transfer channels (Figure 2e,f). The effective doping of iron and nitrogen is evidenced by the elemental mapping shown in Figure 2g, which demonstrates a uniform distribution of carbon, oxygen, nitrogen, and iron throughout the carbon framework.

#### 3.1.2. Structural Analysis

Figure 3a,b display the XRD patterns of the WSC-Fe/N-T samples before and after HCl washing. The XRD pattern of the WSC-Fe/N-800 sample showed distinct diffraction peaks at 2θ = 30.1° and 43.1°, corresponding to the lattice planes of Fe_3_O_4_ [23]. This indicates that the FeO(OH) deposits on the carbon precursor were converted to Fe_3_O_4_ at temperatures below 800 °C. For the WSC-Fe/N-900 and WSC-Fe/N-1000 samples, the intensity of the Fe_3_O_4_ diffraction peaks decreased, suggesting that Fe_3_O_4_ was reduced to other amorphous substances as the carbonization temperature increased. After HCl washing, the Fe_3_O_4_ diffraction peaks disappeared, indicating that acid washing effectively removed Fe_3_O_4_ from the samples. In addition, the XRD patterns of the WSC-Fe/N-T samples revealed two prominent peaks at 25° and 43°. These peaks correspond to the (002) and (101) planes of carbon. The peak at 25° was broader and weaker, indicative of an amorphous structure, whereas the peak at 43° was also weak, suggesting a slight change in the graphitic structure of the prepared materials. All samples exhibited disordered porous carbon structures [24]. The XRD patterns revealed that the WSC-Fe/N-900 sample exhibited significantly higher intensity in the low-angle scattering region (Figure 3b) compared with the samples treated at other temperatures. This observation indicates that the sample possesses well-developed nanoporosity, which enhances both capacitance performance and electrolyte ion adsorption at the conductor surface [25].

The pore size and specific surface area of carbon conductor materials are crucial factors affecting their energy storage performance, predominantly determining the ion adsorption and desorption procedures at the electrode–electrolyte surface. To assess the porosity of the sample, nitrogen adsorption–desorption isotherms were utilized. BET analysis (Figure 3c) revealed that all specimens presented an amalgamation of Type IV and Type I isotherm behaviors. The WSC-Fe/N-900 pore size distribution curve revealed that, as predicted, micropores and mesopores predominated (Figure 3d). A significant increase in the isotherm at low relative pressures (P/P_0_ < 0.01) indicates a substantial amount of micropores in the sample. Additionally, the hysteresis loop observed at elevated relative pressures (P/P_0_ = 0.4–1.0) confirmed the presence of mesopores within the sample matrix. Following the increase in pyrolysis temperature from 800 °C to 900 °C, the average pore width decreased from 2.48 to 1.58 nm and the total pore volume increased significantly, along with a notable rise in micropore volume. However, when the temperature was elevated to 1000 °C, the average pore width increased to 3.54 nm and the volume of mesopores decreased significantly. These observations suggest that the higher pyrolysis temperature caused the structural collapse of the pore network. Furthermore, the micropore volume of WSC-Fe/N-900 was markedly higher than those of WSC-900 and WSC-N-900, highlighting the role of Fe-Nx sites in promoting additional micropore formation. As depicted in Appendix A, the specific surface area of the WSC-N-900 specimen is more elevated than that of WSC-900, which can be attributed to the formation of micropores via the etching action of NH3 produced from the high-temperature decomposition of NH4Cl. Moreover, the thermal degradation temperature has a significant impact on the specific surface area of the specimens. It increases from 2101.4 m^2^/g for WSC-Fe/N-800 to 2576.6 m^2^/g for WSC-Fe/N-900, and then decreases to 2279.4 m^2^/g for WSC-Fe/N-1000. The reduction in specific surface area at higher pyrolysis temperatures is due to the collapse of some pore structures during the pyrolysis process [26].

(The specific surface area of all samples of S5 has been shown in the Appendix A.)

All samples exhibit elements in their respective chemical states, as demonstrated by the XPS spectra presented in Figure 4a. The detailed XPS spectra for the C 1s, O 1s, N 1s, and Fe 2p of the WSC-Fe/N-900 sample are shown in Figure 4. As illustrated in Figure 4b, the C 1s spectrum exhibits prominent summits at 284.8 eV (C-C/C-H), 285.5 eV (C-N), 286.5 eV (C-O), and 288.5 eV (C=O). These peaks indicate successful nitrogen doping into the carbon matrix [27]. The O 1s spectrum (Figure 4c) has peaks at 530 eV for oxide, 531.5 eV for hydroxyl, and 533 eV for hydrated. This suggests that surface oxidation and hydroxyl groups may have an impact on the stability and electrochemical effectiveness of the material. The Fe 2p spectra displayed peaks at 711.4 eV (Fe 2p3/2) and 724.4 eV (Fe 2p1/2) (Figure 4d), indicating the presence of Fe-Nx coordination and confirming the successful formation of Fe-N bonds. None of the samples exhibited indications of metallic iron [28]. Figure 4e displays the N 1s spectrum, which features peaks at 398.1 eV, 399.5 eV, 400.4 eV, 400.9 eV, and 401.7 eV. These peaks correspond to nitrogen oxides, graphitic nitrogen, pyridinic nitrogen, Fe-Nx, and pyrrolic nitrogen, respectively. (The XPS spectra of WSC-Fe/N-800 and WSC-Fe/N-1000 have been presented in the Appendix A, which are shown in Appendix A and Appendix A, respectively).

Appendix A summarizes the nitrogen concentration for all samples across different configurations. Among the Fe/N-doped samples, the WSC-Fe/N-900 sample exhibited the highest total concentrations of nitrogen (4.61 at.%) and iron (0.82 at.%). Inductively coupled plasma optical emission spectroscopy further revealed an iron content of approximately 1.7 wt.% in WSC-Fe/N-900. The WSC-Fe/N-900 sample also exhibited the highest levels of Fe-Nx, pyridinic nitrogen, graphitic nitrogen, pyrrolic nitrogen, and nitrogen oxides, with concentrations of 1.71,1.26, 0.71, 0.71, and 0.21 at.%, respectively. Notably, Fe-Nx has a beneficial effect on the supercapacitor capacitive performance, and pyridinic nitrogen and pyrrolic nitrogen increase capacitance through pseudocapacitance. Graphitic nitrogen improves capacitance by altering the charge density and reducing electron transfer resistance [29].

Figure 4f shows the Raman spectra, exhibiting characteristic peaks at 1605 cm^−^^1^ (G band) and 1340 cm^−^^1^ (D band). The degree of defects in the samples was assessed using the D/G peak intensity ratio (I_D_/I_G_). The I_D_/I_G_ ratio for WSC-Fe/N-900 was 0.91, lower than those for WSC-Fe/N-800 and WSC-Fe/N-1000. This is consistent with the partial collapse of the porous structure in WSC-Fe/N-1000 and shows a higher degree of graphitization and lower defect density in WSC-Fe/N-900 (Figure 4f). This suggests that an excessive carbonization temperature lowers the degree of graphitization [30].

In summary, the capacitance performance of the WSC-Fe/N-900 electrode was significantly enhanced by the synergistic effect of Fe-N-C. This observation suggests that co-doping with iron and nitrogen can optimize the energy storage capacity of the supercapacitor electrode material and markedly enhance its electrochemical performance [31]. The Raman spectroscopy analysis revealed that carbonization temperature had a significant impact on the defect density and graphitization degree of WSC-Fe/N-T, with WSC-Fe/N-900 exhibiting the optimal defect structure at 900 °C, which enhanced its electrochemical performance [32].

### 3.2. Electrochemical Performance of Supercapacitors

The electrochemical effectiveness of all specimens was assessed using a three-electrode system with a 6 M KOH electrolyte solution. The CV curves of the samples (Figure 5a) display an approximately rectangular shape, indicative of typical electric double-layer capacitance (EDLC). The greatest area was displayed by WSC-Fe/N-900, indicating greater charge storage capacity. The CV curves of the WSC-Fe/N-T series also showed a faint broad peak, suggesting the presence of extra pseudocapacitance from redox processes involving nitrogen and oxygen heteroatoms as well as pseudocapacitance linked to Fe-Nx structures [33]. At scan rates as high as 100 mV s^−^^1^, the CV trends of WSC-Fe/N-900 (Figure 5c) remained largely undistorted and stable across the range of 5 to 100 mV s^−^^1^, demonstrating excellent rate performance and steadiness.

All the GCD trends of the electrode samples (Figure 5d) created extremely symmetrical isosceles triangles, which are indicative of normal EDLC function. The specific capacitance of WSC-Fe/N-900 was determined using Equation (S1), yielding a value of 376 F g^−^^1^. This value aligns with the maximum curve area observed in the CV curve, confirming the reliability of the maximum capacitance [34]. At electrical densities extending from 0.5 to 10 A g^−^^1^, the GCD curves of WSC-Fe/N-900 (Figure 5e) exhibit symmetrical, nearly isosceles triangles, indicating exceptional electrochemical reversibility. At a current density of 0.5 A g^−^^1^, the specific capacitance reached 400.5 F g^−^^1^. As the current density increased, ion diffusion within the electrode material became restricted, resulting in reduced electrochemical activity. Consequently, the specific capacitance decreased at higher current densities due to the limited ability of the electrolyte to fully penetrate the electrode surface during rapid charge–discharge cycles. Nevertheless, even at a current density of 10 A g^−^^1^, a superior specific capacitance of 308 F g^−^^1^ was maintained, with a retention of 76.9%. This demonstrates exceptional charging/discharging speed and surpasses that of recently reported carbon compounds derived from biomass (Appendix A) [16,33,35,36,37,38,39,40,41,42]. Conversely, the retention rates of the WSC-Fe/N-800 and WSC-Fe/N-1000 electrodes were 50% and 65%, respectively. Figure 5f shows the computed capacitance values of the electrodes fabricated from each sample at various current densities. The superior capacitance performance of WSC-Fe/N-900 can be attributed to the advantageous effects of the Fe-Nx structure on capacitance, joined with its excellent specific surface area and extensive mesoporous structure [43].

This study utilized electrochemical impedance spectroscopy to investigate the charge transfer rates and ion transport in the samples. Figure 5f displays the Nyquist graphs of all specimens assessed at voltage under open-circuit conditions. The diameter of the semicircle in the Nyquist plot denotes the charge transfer resistance (Rct), while the x-axis intercept shows the equivalent series resistance (Rs). Lower resistance to ion diffusion is indicated by a steeper linear part. The steep linear curve of WSC-Fe/N-900 indicates nearly perfect capacitance performance. Furthermore, WSC-Fe/N-900 exhibited the lowest Rs, indicating superior surface accessibility to electrolyte ions and the lowest internal and interfacial resistance among all the samples. The equivalent circuit model, illustrated in the inset of Figure 5f, was employed to fit the Nyquist curve. The WSC-Fe/N-900 electrode exhibited improved conductivity and better charge transfer, as indicated by its lowest Rct, which is strongly correlated with the electrode’s capacity for charge transfer. In addition to its higher degree of graphitization, WSC-Fe/N-900 exhibited low resistance due to its abundant microporous and mesoporous structure. This structure facilitates shortened ion diffusion paths and reduced charge transfer resistance [44].

The rapid charge–discharge kinetics of WSC-Fe/N-900 were analyzed using Equations (S2) and (S3). Figure 6a illustrates the linear relationship between log(i) and log(v) at various potentials during both the charging and discharging processes. The b-value is typically used to evaluate the redox reaction kinetics; a fast surface-controlled process is indicated by a b-value near to one [30]. The b-values varied with voltage between 0.92 and 0.96, indicating that a surface-controlled mechanism dominated the process and that ultrafast reaction kinetics were observed [45]. A quantitative assessment of capacitance contributions at various scan rates was performed using Equation (S4). WSC-Fe/N-900 exhibited a surface-controlled capacitance contribution of up to 84.1% at a sweep speed of 50 mV s^−^^1^ (Figure 6b). The capacitance contributions at different scan rates are summarized in Figure 6c, which shows that surface-controlled capacitance contributed more as the sweep speed raised, reaching 91.3% at a sweep speed of 100 mV s^−^^1^. This efficiency is primarily ascribed to the well-suited pore structure and high accessible surface area of WSC-Fe/N-900, which contributes to significant EDLC. A surface abundant in micropores and mesopores provides an optimal environment for electrolyte ion storage, facilitating short-distance ion diffusion to the internal surfaces and thereby enhancing electron transport [46,47,48].

The electrochemical performance of the symmetric supercapacitor, denoted as WSC-Fe/N-900-SC, was evaluated using WSC-Fe/N-900 for both the anode and cathode and a 6 M KOH mixture as the conductors. Figure 7a presents the CV curves of WSC-Fe/N-900-SC at sweep speed ranging from 5 to 100 mV s^−^^1^. Excellent EDLC features were demonstrated as the quasi-rectangular form was maintained with increasing scan rate, even up to 100 mV s^−^^1^. Figure 7b shows the GCD curves of WSC-Fe/N-900-SC at different current densities. Each curve exhibited a triangular shape, with a specific capacitance of 66.2 F g^−^^1^ (derived from Equation (S5)) at a current density of 0.5 A g^−^^1^. Even at a superior current density of 10 A g^−^^1^, the specific capacitance remained at 50.3 F g^−^^1^, corresponding to a capacitance retention of 76%. To assess the practical effectiveness of the supercapacitor, the energy and power densities of WSC-Fe/N-900-SC were calculated using Equations (S6) and (S7). WSC-Fe/N-900-SC achieved a superior energy density of 9.2 Wh kg^−^^1^ at a power density of 250 W kg^−^^1^ and a superior power density of 6.5 kW kg^−^^1^ at 5000 W kg^−^^1^ (Figure 7c). These results indicate a significant advantage over other biomass-derived supercapacitors [37,38,45,46,49,50,51,52,53]. Figure 7d illustrates the cycle stability of the WSC-Fe/N-900-SC supercapacitor tested at a current density of 10 A g^−^^1^. The supercapacitor exhibited remarkable electrochemical stability and significant recoverability, attaining a capacitance retention of 93% and a Coulombic efficiency of 98.9% after 10,000 cycles.

## 4. Conclusions

This study successfully synthesized a novel Fe/N co-doped hierarchical porous carbon material, designated WSC-Fe/N-900, using KOH and FeCl_3_ activation and low-cost wheat straw as the precursor. This material has demonstrated suitability for applications in the semiconductor energy storage industry. Structural characterization and chemical composition analysis demonstrated that WSC-Fe/N-900 possesses a uniformly dispersed Fe-Nx structure, a high specific surface area, extensive atom doping, and a rich network of micropores and mesopores, ensuring consistent and reliable material properties. Its excellent electrochemical performance is characterized by a specific capacitance of 400.5 F g^−^^1^ at a current density of 0.5 A g^−^^1^ and a specific capacitance of 308 F g^−^^1^ even at 10 A g^−^^1^, resulting in a capacitance retention rate of 76.9% under these conditions. Furthermore, the symmetric supercapacitor constructed from this material exhibits a superior energy density of 9.2 Wh kg^−^^1^ at a power density of 250 W kg^−^^1^, along with an impressive power density of 6.5 Wh kg^−^^1^ at 5000 W kg^−^^1^. After 10,000 cycles, it retains a Coulombic efficiency of 98.9% and a capacitance retention rate of 93%, indicating superior reversibility, electrochemical stability, and resistance to aging effects. These results demonstrate that the material maintains structural integrity and stable performance over long-term use.

The methodology employed in this study scientifically transforms agricultural waste, thereby mitigating problems associated with the overuse of such waste. Our study advances energy storage technologies and provides a novel approach for achieving ecological sustainability.

## Figures and Tables

**Figure 1 nanomaterials-14-01692-f001:**
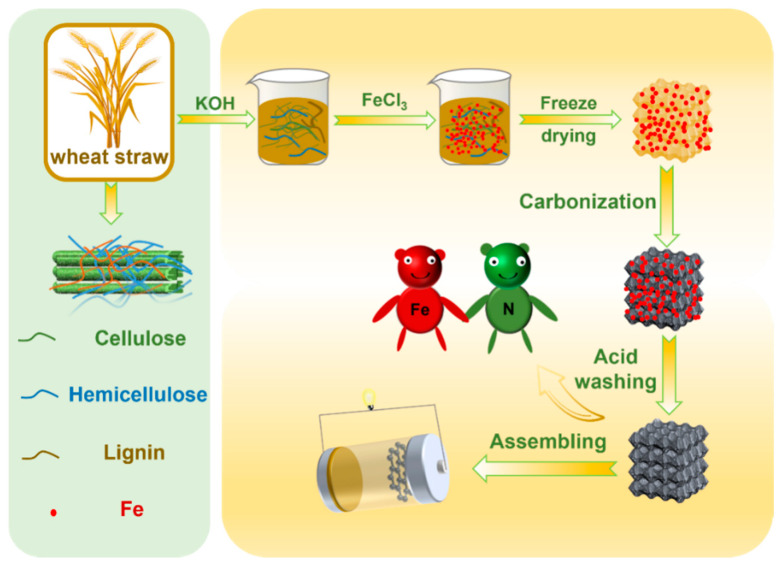
Procedure for the preparation of Fe/N co-doped porous carbon materials from wheat straw.

**Figure 2 nanomaterials-14-01692-f002:**
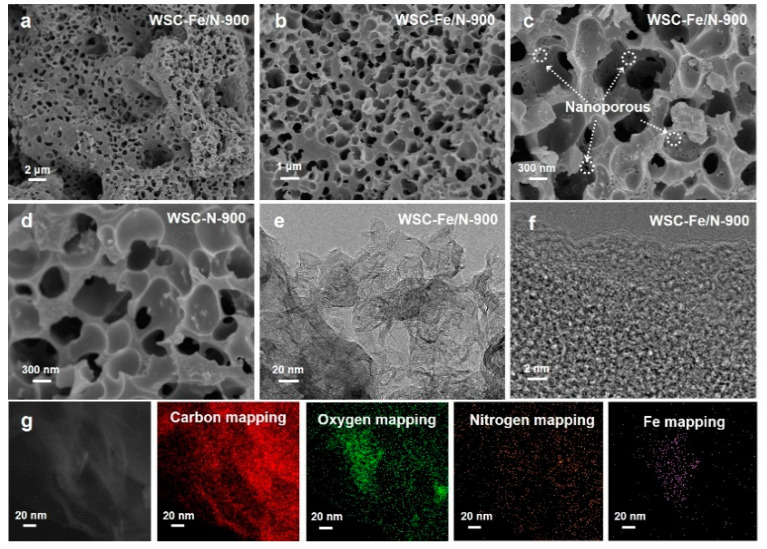
Scanning electron microscope (SEM) images of (**a**–**c**) wheat straw treated with Fe/N and heated to 900 °C (WSC-Fe/N-900) and (**d**) wheat straw treated with Fe without doping. Transmission electron microscope images of (**e**,**f**) WSC-Fe/N-900. (**g**) Elemental distribution in the samples: C (red), O (green), N (orange), and Fe (purple).

**Figure 3 nanomaterials-14-01692-f003:**
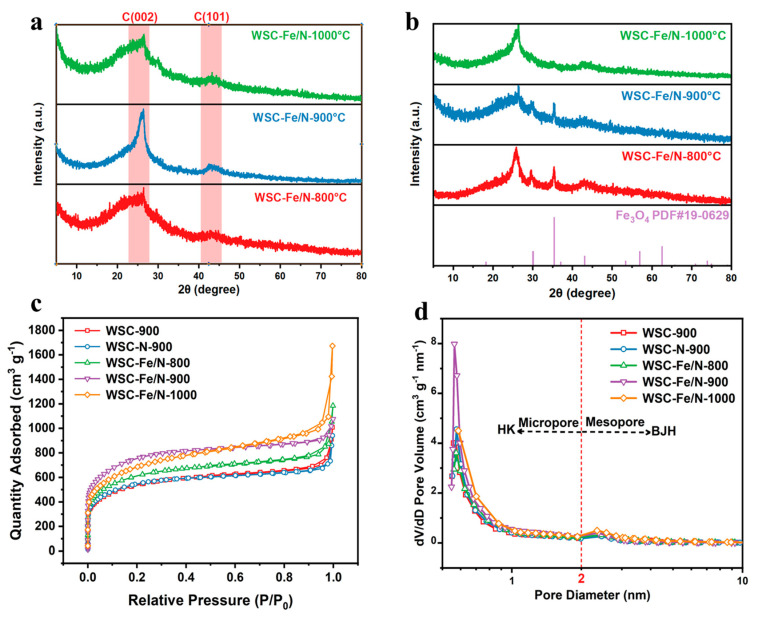
XRD patterns of WSC-Fe/N-800, WSC-Fe/N-900, and WSC-Fe/N-1000 (**a**) before and (**b**) after acid washing. (**c**) N adsorption–desorption isotherms of different control samples. (**d**) Pore size distribution of different control samples calculated.

**Figure 4 nanomaterials-14-01692-f004:**
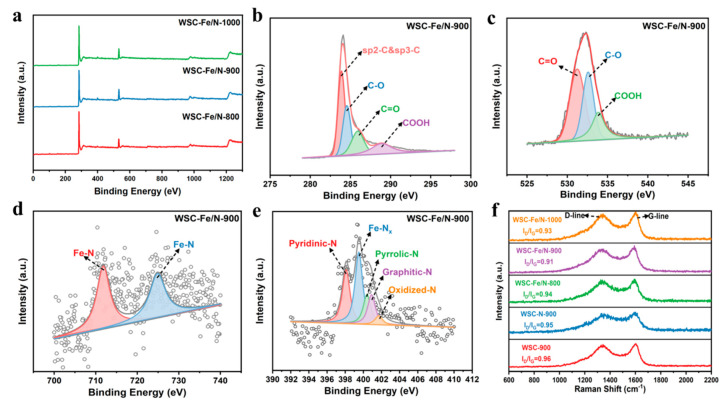
XPS spectra of (**a**) WSC-Fe/N-800, WSC-Fe/N-900, and WSC-Fe/N-1000; (**b**) C 1s, (**c**) O 1s, (**d**) N 1s, and (**e**) Fe 2p for WSC-Fe/N-900. (**f**) Raman spectra of different samples.

**Figure 5 nanomaterials-14-01692-f005:**
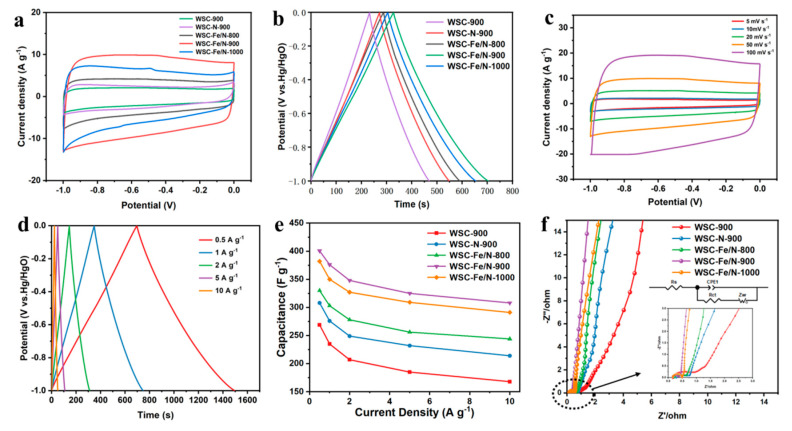
Cyclic voltammetry (CV) curves of (**a**) all comparison samples and WSC-Fe/N-900, (**b**) WSC-Fe/N-T, and (**c**) WSC-Fe/N-900 at various scan rates. Galvanostatic charge–discharge (GCD) curves of (**d**) all comparison samples and WSC-Fe/N-900 and (**e**) specific capacitance of WSC-Fe/N-T (T = 800, 900 and 1000 °C) at different current densities. (**f**) Specific capacitance values of all specimens at different current densities.

**Figure 6 nanomaterials-14-01692-f006:**
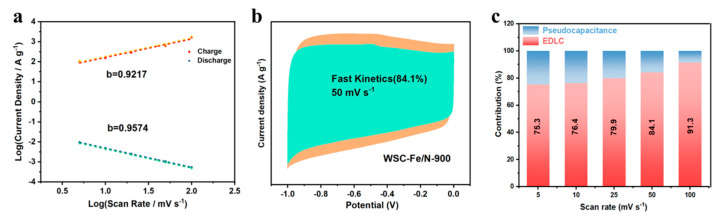
(**a**) Linear relationship between current density and scan rate at various potentials during charge and discharge of WSC-Fe/N-900. Contribution of capacitance at (**b**) 50 mV s^−^^1^ and (**c**) different scan rates for WSC-Fe/N-900.

**Figure 7 nanomaterials-14-01692-f007:**
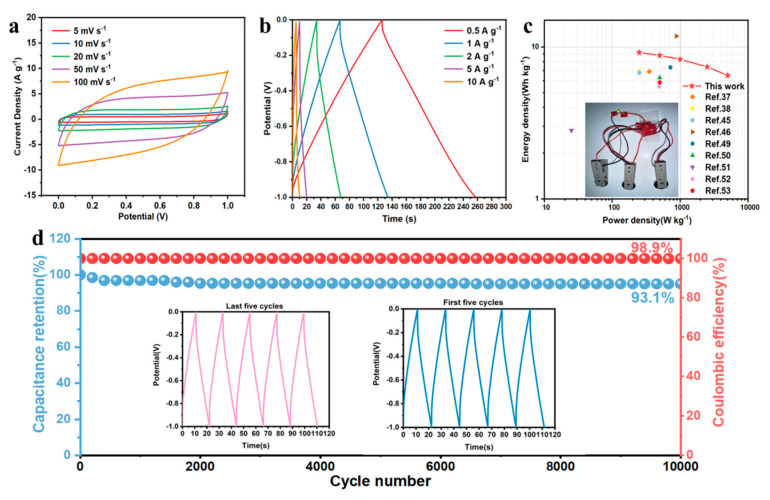
(**a**) CV curves of the WSC-Fe/N-900 supercapacitor (WSC-Fe/N-900-SC) at different scan rates. (**b**) GCD curves of WSC-Fe/N-900-SC at various current densities. (**c**) Ragone plot of WSC-Fe/N-900-SC. (**d**) Stability of WSC-Fe/N-900-SC (GCD curves at the beginning and end of the cycle stability test are shown in the inset).

## Data Availability

The original contributions presented in the study are included in the article/Appendix A, further inquiries can be directed to the corresponding author/s.

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
