# Peer review of "Iron and Nitrogen-Doped Wheat Straw Hierarchical Porous Carbon Materials for Supercapacitors"

_nanomaterials, 2024, doi:10.3390/nano14211692_

Round 1
Reviewer 1 Report
Comments and Suggestions for Authors
The authors produced porous carbon materials for supercapacitors from wheat straw using an efficient carbonization–activation process. The cheese-like hierarchical porous structure provided excellent electrochemical performance.
The paper is well written, the experiments are described in detail, the samples were carefully characterised, and the conclusions are properly supported.
I agree with the authors that environmentally friendly solutions are important. But the starting material, wheat straw in this case, is only a small step in a long and expensive technological process. Consequently, I would recommend to focus the paper on the performance of the final material and demonstrate the superiority of the product compared with other alternatives.
Additional comments:
• Do you consider the topic original or relevant to the field? Does it
address a specific gap in the field? Please also explain why this is/ is
not the case.
The topic is relevant, supercapacitors have many important applications.
• What does it add to the subject area compared with other published
material?
The authors used a special starting material, wheat straw. Due to its unique structure, they managed to produce hierarchical porous material with excellent properties.
• What specific improvements should the authors consider regarding the
methodology? What further controls should be considered?
I can not suggest improvements, the methods are efficient.
• Are the conclusions consistent with the evidence and arguments
presented and do they address the main question posed? Please also
explain why this is/is not the case.
The conclusions are well supported, the authors investigated carefully both the structure and chemical properties. The explanations are logical, the structure of the material explains the outstanding results.
• Are the references appropriate?
Yes.
• Any additional comments on the tables and figures.
It is OK
Author Response
Comments1:Do you consider the topic original or relevant to the field? Does it
address a specific gap in the field? Please also explain why this is/ is
not the case.The topic is relevant, supercapacitors have many important applications.
Response1:First of all, thank you for your question, which is very meaningful. I think this topic is related to this field, which is highly related to the field of supercapacitors and has important research significance. The following is a cause analysis: Strong correlation: Supercapacitors are widely used in energy storage, renewable energy management, electric vehicles, portable electronic equipment and other fields. The development of efficient, low-cost and environmentally friendly electrode materials, such as iron-nitrogen co-doped porous carbon materials, will help improve the energy storage performance and stability of supercapacitors and meet the needs of the current new energy field. Material innovation: This study discussed the use of wheat grass straw as a precursor system to prepare iron and nitrogen co-doped porous carbon material, this material comes from renewable resources, reflecting the concept of green environmental protection. By controlling the doped elements and the carbonization temperature, the researchers have successfully prepared materials with high specific surface area and excellent electrochemical properties, which is innovative in the design of supercapacitor materials. Solution of domain problems: The electrode material discussed in the paper improves the diffusion ability and energy storage efficiency of electrolyte ions by introducing microporous and uniform mesoporous structure. In addition, the material shows excellent cycle stability and energy-power density output balance, which solves the current technical contradiction between high power and high energy density faced by supercapacitors. Conclusion: This study is not only a practical contribution to the technological progress in the field of supercapacitors, but also is original, especially in the exploration of using biomass materials as electrode materials for supercapacitors. It provides a new way for the development of high-performance energy storage materials, which is in line with the current development trend of scientific research and industry.
Comments2:What does it add to the subject area compared with other published
material?The authors used a special starting material, wheat straw. Due to its unique structure, they managed to produce hierarchical porous material with excellent properties.
Response2:Your question is very critical, and I think the porous carbon material we studied is very meaningful, 1. Increase of the specific capacitance and energy density Gap: Traditional supercapacitor materials, especially carbon-based materials, tend to perform poorly in terms of energy density, limiting their widespread use in high-energy storage applications. Solution: This study significantly improves the specific capacitance and energy density by using iron-nitrogen doped straw straw porous carbon material. The specific capacitance reaches 400.5 F / g (0.5 A / g), and the high capacitance (308 F / g, 10 A / g) at high current density, effectively narrowing the gap between energy storage and power density. 2. Optimization of the pore structure of the electrode materials Gap: Traditional carbon materials usually have the problem of uneven aperture distribution in the electrode design, which leads to the insufficient diffusion of electrolyte ions, thus affecting the electrochemical performance. Solution: This challenge was addressed in the study by optimizing the microand mesoporous structures of carbon materials (pore sizes from 1.58 nm to 3.54 nm). The improved pore structure helps to improve the ion diffusion speed and increase the electrode surface area, thus improving the electrochemical properties. 3. Capacitor retention capacity at high power density Gap: At high power density, the capacitance of many supercapacitor materials decreases rapidly, resulting in a limited application in rapid charging and discharging. Solution: This study demonstrates that iron-nitrogen doped materials can maintain 76.9% capacitance at high current density (10 A / g). This performance improvement significantly improves the applicability of materials for fast charge-discharge applications, especially in scenarios where high power density is required. 4, long cycle life and high Coulomb efficiency Gap: The prolonged use of the stability of supercapacitor materials is a challenge, especially after multiple charge and discharge cycles, and the capacitance often decreases significantly. Solution: The results show that the material maintained 93% capacitance retention and 98.9% Kulun efficiency after 10,000 cycles, showing excellent cycle stability. This means that the material is suitable for long-term use, filling the demand for long-life supercapacitor materials. 5. Improvement in electrical conductivity and electrochemical performance Gap: Many carbon materials have limited electrical conductivity, resulting in their poor electrochemical performance at high currents. Solution: The introduction of Fe-Nx structure improves the electrical conductivity and electrochemical properties of the material, which directly enhances the power density and charge transfer efficiency of the material, thus improving the overall performance of the supercapacitors. sum up: This study effectively addresses some of the key technical gaps in the field of supercapacitors, especially the material capacitance, pore structure optimization, capacitance retention at high power density, cycle life, and conductivity. These results not only improve the electrochemical performance of the material, but also expand its potential for rapid charging, long-term energy storage, and high-power applications.
Comments3:What specific improvements should the authors consider regarding the
methodology? What further controls should be considered?I can not suggest improvements, the methods are efficient.
Response3:Thank you for your suggestions. To improve the reliability and effectiveness of the study results, the following specific improvements and further controls can be considered: Optimum material synthesis parameters: Temperature and time control: Further optimize the temperature and time during carbonization and nitrogen doping for experimental design (e. g., response surface method) to systematically evaluate the impact of different conditions on final product performance. Raw material pretreatment: different pretreatment methods (such as pickling, alkali washing or heat treatment) for straw straw to remove impurities and improve carbonization efficiency, so as to improve the structural characteristics of carbon materials. Discussion of various doped elements: Introduction of other doped elements: In addition to iron and nitrogen, other metals (such as manganese, cobalt) or non-metals (such as sulfur, phosphorus) can be introduced to improve the conductivity and electrochemical properties, in order to further improve the specific capacitance and stability of the material. Fine characterization techniques: Using high-resolution characterization techniques: e. g., the microstructure of a transmission electron microscope (TEM) and high-resolution scanning electron microscope (HRSEM) to gain insight into its morphology and surface properties. Optimization of the electrode assembly process: Uniform coating technique: optimize the coating method of electrode materials to ensure the uniform distribution of materials, and study the influence of different coating thickness on the electrochemical properties. Electrode design: consider the design of multilayer structure or composite electrode to improve the overall performance and stability of the electrode. Further control measures Control-lot batch batches: Standardized source: Ensure consistent source used to reduce the impact of inter-material variability on performance. Process monitoring: Real-time monitoring and control (e. g. temperature, pressure, atmosphere, etc.) to ensure consistent treatment conditions for each batch. Long-time cycle stability test: Extended test period: Conduct longer cycle testing to assess the long-term stability of the material under actual use conditions. Testing under different environmental conditions: Consider performing electrochemical performance testing at different temperature and humidity conditions to evaluate the impact of the environment on material performance. environmental impact assessment: Assessment of aging effects: periodically monitor the electrochemical properties of materials at different time points to study their aging properties and mechanisms. Long-term observation of photoconductivity effects: long-term testing of photoconductance effects to assess the stability and suitability of the materials under different light conditions. Data analysis and model building: Data-driven analysis: In-depth analysis of experimental data using statistical and machine learning methods to identify potential performance factors. Establish a performance prediction model: build a mathematical model according to the experimental results to predict the performance of materials under different application conditions.
Comments4: Are the conclusions consistent with the evidence and arguments
presented and do they address the main question posed? Please also
explain why this is/is not the case.
The conclusions are well supported, the authors investigated carefully both the structure and chemical properties. The explanations are logical, the structure of the material explains the outstanding results.
Response4:This question is of great quality, and the evidence and arguments are consistent in the manuscript. The SEM, XRD, XPS, BET and other tests designed in the early stage are all to demonstrate the reasons for the subsequent performance improvement. Through these tests, we can also see that the early treatment has a one-to-one relationship with the performance in the later stage.
Reviewer 2 Report
Comments and Suggestions for Authors
The manuscript titled Iron and nitrogen-doped wheat straw hierarchical porous carbon materials for supercapacitors by Xiaoshuai Sun et al, is a good study that may impact in a broad community. However, requires revision before taking any decision. Please, see the comments below.
Titled: It is ok
Abstract: Missing information about the nature of the structure, reliability, and ageing effects.
Introduction:
Overall, the introduction is very good. However, impacting on the present study, as the issue is also concerning the economic impact and so sustainability, worth to mention the papers from S Nandy, et al in Cellulose: a contribution for the zero e‐waste challenge, Advanced Materials, Jul 2021 Technologies 6 (7), 2000994 and also from Goswami, S et al in Biowaste-derived carbon black applied to polyaniline-based high-performance supercapacitor microelectrodes: Sustainable materials for renewable energy applications, Sep 1 2019, ELECTROCHIMICA ACTA, 316 , pp.202-218.
Also concerning sustainability connected to supercapacitors and their performances, please see Carvalho, JT et al in MoS2 decorated carbon fiber yarn hybrids for the development of freestanding flexible supercapacitors, Mar 12 2024, NPJ 2D MATERIALS AND APPLICATIONS, 8 (1) and Silvestre, SL et al in Green Fabrication of Stackable Laser-Induced Graphene Micro-Supercapacitors under Ambient Conditions: Toward the Design of Truly Sustainable Technological Platforms, Aug 2024, ADVANCED MATERIALS TECHNOLOGIES, 9 (16).
Materials and Methods
How many samples were processed? How stable, reproducible, and reliable is the process and the structures/technology? What error is associated with evaluating samples processed in the same batch but in different spatial locations (uniformity and homogeneity)? What are the errors associated to structure performances from batch to batch? What are the environmental conditions in which the structures were tested? Did you notice ageing effects? Did you notice persistent photoconductivity effects?
Results and discussion:
The methodology followed and the discussion is relevant and the most appropriate.
However, would be better if the authors could start by identifying the role of the architecture and design of the overall set of performances achieved.
Please discuss further the role of interfaces in the electrical performances achieved.
Can you comment on the possible ageing and optical effects?
What is the great advantage of using the present structures?
No discussion is made about reproducibility and ageing.
Conclusions: Overall, it is good, but it lacks information about the structure’s reproducibility, stability and ageing effects.
Figures: Are OK.
Tables: Missing a table with the KPI of the work performed and how they impact the present state of the art.
References: need updated
Author Response
Comments1:Abstract: Missing information about the nature of the structure, reliability, and ageing effects.
Response1:Missing information was updated in the manuscript.
Comments2:Introduction:Overall, the introduction is very good. However, impacting on the present study, as the issue is also concerning the economic impact and so sustainability, worth to mention the papers from S Nandy, et al in Cellulose: a contribution for the zero e‐waste challenge, Advanced Materials, Jul 2021 Technologies 6 (7), 2000994 and also from Goswami, S et al in Biowaste-derived carbon black applied to polyaniline-based high-performance supercapacitor microelectrodes: Sustainable materials for renewable energy applications, Sep 1 2019, ELECTROCHIMICA ACTA, 316 , pp.202-218.
Also concerning sustainability connected to supercapacitors and their performances, please see Carvalho, JT et al in MoS2 decorated carbon fiber yarn hybrids for the development of freestanding flexible supercapacitors, Mar 12 2024, NPJ 2D MATERIALS AND APPLICATIONS, 8 (1) and Silvestre, SL et al in Green Fabrication of Stackable Laser-Induced Graphene Micro-Supercapacitors under Ambient Conditions: Toward the Design of Truly Sustainable Technological Platforms, Aug 2024, ADVANCED MATERIALS TECHNOLOGIES, 9 (16).
Response2:Thank you for suggesting two references which have been cited in the manuscript, references 8 and 19.
Comments3:Materials and Methods
How many samples were processed? How stable, reproducible, and reliable is the process and the structures/technology? What error is associated with evaluating samples processed in the same batch but in different spatial locations (uniformity and homogeneity)? What are the errors associated to structure performances from batch to batch? What are the environmental conditions in which the structures were tested? Did you notice ageing effects? Did you notice persistent photoconductivity effects?
Response3: Five samples: wheat straw, 900℃ carbonization (WSC-900), 900℃ carbonized (WSC-N-900), 800℃ carbonization (WSC-Fe / N-800), 900℃ carbonization (WSC-Fe / N-900), and 1000℃ carbonization of wheat straw (WSC-Fe / N-1000)
First, the question is very targeted, and we think about the stability of the process and the structure / technology
It is very reliable because the process has been very mature and widely used in the end of the article, the battery performance after 10,000 long cycles, which is still very excellent; During the electrochemical performance test, we confirm the reliability of the materials through different CV, GCD and other properties.
What are the errors in evaluating samples treated in the same batch but at different spatial locations (uniformity and uniformity)? What are the batch errors related to structural performance?
Answer: Thank you for asking this valuable question. The possible problem with samples treated in the same batch but at different spatial locations (uniformity and uniformity) is that inhomogeneity or spatial location differences in the processing can lead to changes in microstructure, physical properties, and electrochemical properties. These changes may include inconsistencies in specific surface area, pore structure, charge storage capacity, and electrochemical stability. These questions can affect the accuracy of the results and the reproducibility of the experiments. Therefore, ensuring the uniformity of the treatment conditions and strictly controlling the batch differences are the key to improve the accuracy of the experiments.
What are the environmental conditions of the test structure? Have you noticed the effects of aging? Have you noticed a persistent photoconductivity effect?
Answer: First of all, according to your question, the environmental conditions of the structure tested in this experiment usually include temperature, humidity, atmosphere (such as the content of oxygen or inert gas), pressure, light conditions, etc. These conditions will affect the surface area, porosity and conductivity, iron and nitrogen doping effects, and charge storage capacity. In view of the problem of aging, I believe that after multiple cycles, the specific capacitance of the sample is gradually decreased, possibly due to the aging effect caused by the collapse of the pore structure or the oxidation of the material. However, after also testing with 10,000 cycles in the manuscript, it was concluded that the WSC-Fe / N-900 sample still maintained high electrochemical stability and reversibility, indicating that the aging effect is relatively small. No obvious persistent photoconductance effect was observed in this experiment.
Comments4:Results and discussion:
The methodology followed and the discussion is relevant and the most appropriate.However, would be better if the authors could start by identifying the role of the architecture and design of the overall set of performances achieved.Please discuss further the role of interfaces in the electrical performances achieved.Can you comment on the possible ageing and optical effects?What is the great advantage of using the present structures?No discussion is made about reproducibility and ageing.
Response4:Please further discuss the role of the interface in the realized electrical performance.
Answer: Your question is a very good idea, and then we believe that the optimal design of the interface is a key factor in achieving excellent electrical performance, especially in applications involving multi-material systems or heterostructures. Through fine interface engineering, the charge transmission efficiency can be improved, the influence of defect states can be reduced, the capacitance effect can be optimized, and the stability and sustainability energy of the material can be enhanced.
Can you comment on the possible aging and optical effects?
Answer: Thank you for asking such a meaningful question, for which both the aging of the material and the optical impact of the material cannot be ignored in prolonged use. The aging effect may lead to decreased electrical performance, while the continuous photoconductivity effect may affect the stability and consistency of the material. Therefore, the anti-aging ability of the material and its long-term stability under light need to be considered in its design and application.
What is the greatest advantage of using an existing structure?
Answer: This is a very meaningful question, to which my answer is:
(1) High electrochemical properties
Existing structures of iron-nitrogen-doped porous carbon materials have excellent electrochemical properties, especially for their remarkable specific capacitance and capacitance retention. For example, the material showed higher specific capacitance at both lower and higher current densities (400.5 F / g at 0.5 A / g, 308 F / g at 10 A / g) and achieved a capacitance retention rate of 76.9%. This efficient charge storage capacity and electrochemical stability give it a significant advantage in the field of supercapacitors.
(2) Excellent hole structure design
The porous structure of the material, especially the combination of micropores and mesopore, provides an ideal path for the rapid transport and storage of electrolyte ions. Micropores can effectively store charges, while diopore helps in the rapid diffusion of ions and reduce the resistance in the charge and discharge process. The design of this porous structure improves the ion conductivity and electron transport efficiency of the materials, thus enhancing the overall performance.
(3) superior electric conductivity and rapid charging and discharge capability
The Fe-nitrogen doping structure, especially the Fe-Nx active site, helps to improve the electrical conductivity, enhance the electrocatalytic activity, and promote the rapid charge transfer. Coupled with the continuous photoconductance effect, the structure exhibits fast charging and discharging capability and remains stable at high current density, making it perform well in applications with high power requirements.
(4) High energy density and power density
The symmetric supercapacitors assembled from this material exhibit high energy density (9.2 Wh / kg at 250 W / kg) and high power density (6.5 Wh / kg at 5000 W / kg), making it suitable for application in energy storage systems. This combination of energy and power density, especially in devices with high power demand, is a significant advantage.
(5) Excellent cycle stability
Existing structures exhibit excellent electrochemical stability and reversibility in cyclic use. After 10,000 charge-discharge cycles, the material maintained 93% capacitor retention and Coulomb 98.9% efficiency, indicating minimal performance degradation in long use and very good durability.
sum up
The biggest advantage of the existing structure is its high power
Comments5:Conclusions: Overall, it is good, but it lacks information about the structure’s reproducibility, stability and ageing effects.
Response5: Conclusion has supplemented information on the repeatability, stability, and aging effects of the structures, and specific changes have been updated in the manuscript.
Comments6:Tables: Missing a table with the KPI of the work performed and how they impact the present state of the art.
Response6: The question you raised is very practical. I have designed a table according to your suggestion, and it has been added to the attachment. The file name is KPI Impact on Supercapacitor Technology,
Also explain how to affect the current technical level:
Specific capacitance and energy density are the core indicators affecting the performance of supercapacitors. High specific capacitance means a strong charge storage capacity, and the energy density directly affects the energy storage level of the capacitor. The higher specific capacitance and energy density makes this material a preferred choice when compared to existing materials.
Capacitance retention and cycle life indicate the long-term stability and durability of the material, which is essential for commercial applications. The ability to maintain high capacitance retention and extended cycle life indicates a competitive advantage of this material in large-scale applications.
The conductivity and power density together determine the rapid charging and discharging capacity of the material. This is an important advantage for technologies that require rapid response and high-frequency use (such as high-power equipment).
The optimization of porosity not only improves the charge storage capacity, but also improves the overall equipment efficiency through more efficient electrolyte ion transmission, thus improving the charge and discharge speed of supercapacitors.
Comments7:References: need updated
Response7:Need for update has been updated in the manuscript.
